# Aging is associated with increased brain iron through cortex-derived hepcidin expression

**Tatsuya Sato[1], Jason Solomon Shapiro[1], Hsiang-Chun Chang[1], Richard A Miller[2], Hossein Ardehali[1]***

[1]Feinberg Cardiovascular and Renal Research Institute, Northwestern University School of Medicine, Chicago, United States; [2]Department of Pathology, University of Michigan School of Medicine, Ann Arbor, United States

**Abstract** Iron is an essential molecule for biological processes, but its accumulation can lead to oxidative stress and cellular death. Due to its oxidative effects, iron accumulation is implicated in the process of aging and neurodegenerative diseases. However, the mechanism for this increase in iron with aging, and whether this increase is localized to specific cellular compartment(s), are not known. Here, we measured the levels of iron in different tissues of aged mice, and demonstrated that while cytosolic non-heme iron is increased in the liver and muscle tissue, only the aged brain cortex exhibits an increase in both the cytosolic and mitochondrial non-heme iron. This increase in brain iron is associated with elevated levels of local hepcidin mRNA and protein in the brain. We also demonstrate that the increase in hepcidin is associated with increased ubiquitination and reduced levels of the only iron exporter, ferroportin-1 (FPN1). Overall, our studies provide a potential mechanism for iron accumulation in the brain through increased local expression of hepcidin, and subsequent iron accumulation due to decreased iron export. Additionally, our data support that aging is associated with mitochondrial and cytosolic iron accumulation only in the brain and not in other tissues.

**\*For correspondence:**
h-ardehali@northwestern.edu

## Editor's evaluation

The authors present a manuscript addressing an important unmet need, aiming to understand the mechanism of neurodegenerative disease, specifically focused on the role of iron overload in the aging brain, that is of interest to basic iron biologists and neurologists as a Short Report of novel significance. The current study demonstrates increased cytosolic and mitochondrial non-heme iron only in the aging brain, increased cortical hepcidin expression, and decreased levels of FPN1, together supporting a hypothesis that cortical hepcidin sequesters iron in neuronal cells and is associated with aging.

## Introduction

Iron is an essential molecule for almost every organism on earth. It is generally found naturally in two oxidation states, either as the oxidized form of ferric iron ($Fe^{3+}$), or as the reduced form of ferrous iron ($Fe^{2+}$). Ferrous iron is the major form of iron used in mammalian cells, and due to the scarcity of the reduced form of iron on earth, mammalian cells have developed mechanisms for efficient uptake of oxidized iron through its solubilization by acidification, followed by reduction and its cellular transportation. Iron can donate and accept electrons from various substrates due to its unique oxidation-reduction properties, making it an important cofactor in mammalian cells. Iron is also essential for

heme and Fe-S clusters and exerts other biological effects through its role in processes such as demethylation, dehydrogenation, and reduction of sulfur (*Koleini et al., 2021*).

Due to its oxidative effect, iron accumulation is hypothesized to lead to oxidative stress and cellular damage that accelerates the process of aging (*Xu et al., 2012*). Aging has been shown to be associated with iron accumulation in various organs and tissues both in humans and in animal models of aging (*Xu et al., 2012*). However, whether iron accumulation occurs in all organs or is specific to one organ is not known. In the brain, iron accumulation has been demonstrated both in animal models of aging (*Benkovic and Connor, 1993*; *Hahn et al., 2009*; *Roskams and Connor, 1994*) and in humans (*Bartzokis et al., 2010*; *Bartzokis et al., 1994a*; *Bartzokis et al., 1994b*; *Zecca et al., 2001*). Additionally, it is now believed that iron accumulation may be the cause of neuronal cell death in some of neurodegenerative diseases (*Lei et al., 2012*). Despite the importance of iron in brain disorders, the mechanism by which iron accumulates in the brain with aging is not known. Previous studies suggest that iron homeostasis in the brain is dependent on normal expression of genes involved in iron uptake into the cells and its cellular storage and regulation (*Qian and Wang, 1998*). Another study demonstrated that only one protein involved in the iron import pathway, divalent metal transporter (DMT1) was altered in aged brain, while the levels of key proteins involved in iron import and export, transferrin receptor-1 (TfR1), and ferroportin-1 (FPN1) were unchanged (*Lu et al., 2017*). However, the mechanism for the age-related increase in iron accumulation in the brain remains unclear.

In this paper, we assessed the levels of iron in different tissues and cellular compartments with aging in mice. Since it has been reported that age-related brain atrophy, which is associated with age-related cognitive decline, is evident in the brain cortex (*Nyberg et al., 2010*), we selected the cerebral cortex as the brain region in the present study. We demonstrate that the cytosolic non-heme iron is increased in the muscle and liver tissue, while the brain cortex has increased non-heme iron both in the mitochondria and in the cytosol. We also demonstrate that hepcidin levels are upregulated in the brain cortex of aged mice, which is associated with ubiquitination of FPN1 and a reduction in its protein levels. Thus, our studies suggest that local expression of hepcidin in the brain may be a driver in increasing iron levels due to a reduction in iron export by FPN1 degradation.

## Results
### Iron is increased in the brain cortex with aging
To determine whether iron accumulates in tissues with aging, we measured heme and non-heme iron levels in the mitochondria and cytoplasm of liver, skeletal muscle, and brain tissues. These organs were selected since liver is the main iron storage organ, while skeletal muscle is a major iron-consuming organ. Additionally, brain iron accumulation is reported with neurodegenerative diseases. We chose two time points for our studies: 4 months for young mice and 22 months for old mice. The almost 2 years age of the old mice is considered sufficient to result in senescence of some organs (*Flurkey et al., 2007*). Our data demonstrated that cytosolic non-heme iron in the liver was increased with age, whereas liver iron contents in the mitochondria, where iron accumulation is associated with cell dysfunction and cell death, were unchanged with age for both heme and non-heme forms (*Figure 1A and B*). Similarly, we observed an increase in the cytosolic, but not mitochondrial, non-heme iron in the gastrocnemius muscle. We also did not see a change in the cytosolic heme iron; however, we were not able to detect heme iron in the mitochondria of skeletal muscle tissue, possibly due to technical limitations associated with the isolation of intact mitochondria from skeletal muscle tissue (*Figure 1C and D*).

We then studied iron levels in the brain mitochondrial and cytosolic fractions, and noted a significant increase in non-heme iron in both the mitochondria and cytoplasm of dissected brain cortex in older mice (*Figure 1E*). However, there was no increase in heme iron levels in the either compartment in the brain cortex (*Figure 1F*). The purity of the mitochondrial fraction in the brain cortex tissue was confirmed by immunoblotting (*Figure 1G*). Consistent with increased non-heme iron levels, we also observed increased protein levels of ferritin light and heavy chains (*Figure 1H and I*). This increase may be a cellular protective mechanism because free or loosely bound iron in the form of non-heme iron is detrimental to the cells, and iron stored in ferritin is considered redox-inert. Additionally, an elevated level of cellular iron leads to the degradation of IRP2 protein (*Wang et al., 2004*). We

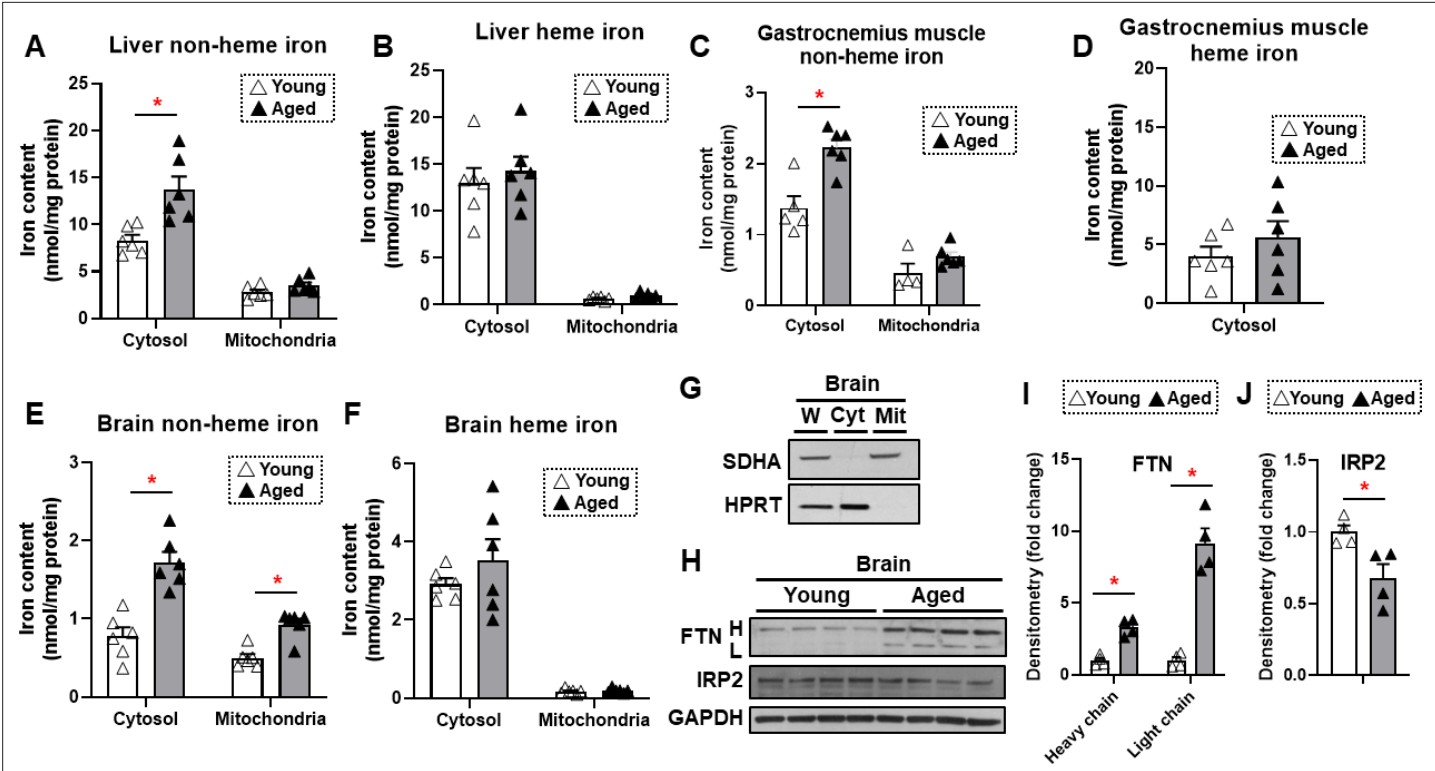

**Figure 1.** Non-heme iron is increased in the cytosol and mitochondria of the brain cortex with aging. Liver non-heme (**A**) and heme (**B**) iron in young (4 months old) and old (22 months old) mice from the cytosol and mitochondria (n=6). Gastrocnemius non-heme (**C**) and heme (**D**) iron in young and old mice from the cytosol and mitochondria (n=6). Outliers in non-heme iron in the cytosol (n=1) and in the mitochondria (n=2) were excluded. Mitochondrial heme iron in gastrocnemius muscle was not detectable in our studies. Brain non-heme (**E**) and heme (**F**) iron in young and old mice from the cytosol and mitochondria (n=6). (**G**) Immunoblots of mitochondrial and cytosolic markers, confirming the purity of mitochondrial isolation in the brain cortex tissue. W = whole cell lysate, Cyt = cytosolic fraction, Mit = mitochondrial fraction, SDHA = Succinate Dehydrogenase Complex Flavoprotein Subunit A, HPRT = hypoxanthine phosphoribosyltransferase. (**H**) Immunoblots of FTN and IRP2 in the brain of young and aged mice (n=4). FTN = ferritin, IRP2 = iron regulatory protein 2, H = heavy chain, L = light chain. Densitometric quantification of light and heavy chain FTN (**I**) and IRP2 (**J**) in the brain of young and aged mice. * $p<0.05$.

The online version of this article includes the following source data for figure 1:

**Source data 1.** Full-length images of immunoblots shown in *Figure 1*.

**Source data 2.** Iron measurement – Original data of iron measurements shown in *Figure 1*.

**Source data 3.** Densitometry – Original data of densitometric quantifications of immunoblots shown in *Figure 1*.

therefore measured IRP2 protein levels in the brain cortex and observed a decrease in IRP2 protein levels in the brain cortex of aged mice (*Figure 1H and J*), consistent with an increase in iron levels with aging.

## Hepcidin levels are increased in the brain with aging

Since non-heme iron is increased in both the cytosolic and mitochondrial compartments of the brain cortex and previous reports suggest a potential role for iron in neurodegenerative diseases associated with aging such as Alzheimer's and Parkinson's disease (*Berg et al., 2001*; *Smith et al., 1997*), we then focused our studies on the mechanism for the increase in iron in the brain tissue. We measured the mRNA levels of all proteins involved in cellular iron homeostasis. We conducted these studies both in the tissues of liver and brain cortex. In the liver, we noted a significant increase in the mRNA levels of *Fpn1*, heme oxygenase 1 (*Hmox1*), ALA dehydrogenase (*Alad*), and bone morphogenetic protein 6 (*Bmp6*), and a significant decrease in the mRNA level of Metalloreductase Six-Transmembrane Epithelial Antigen of Prostate-3 (*Steap3*) (*Figure 2A*). In contrast, we noted a significant increase in the mRNA levels of *TfR2*, mitoferrin 2 (*Mfrn2*), tristetraprolin (*Ttp*), *Hmox1*, *Alad*, and hepcidin (*Hamp1*) in the brain cortex of aged mice (*Figure 2B*). TfR2 does not play a meaningful role in iron uptake compared

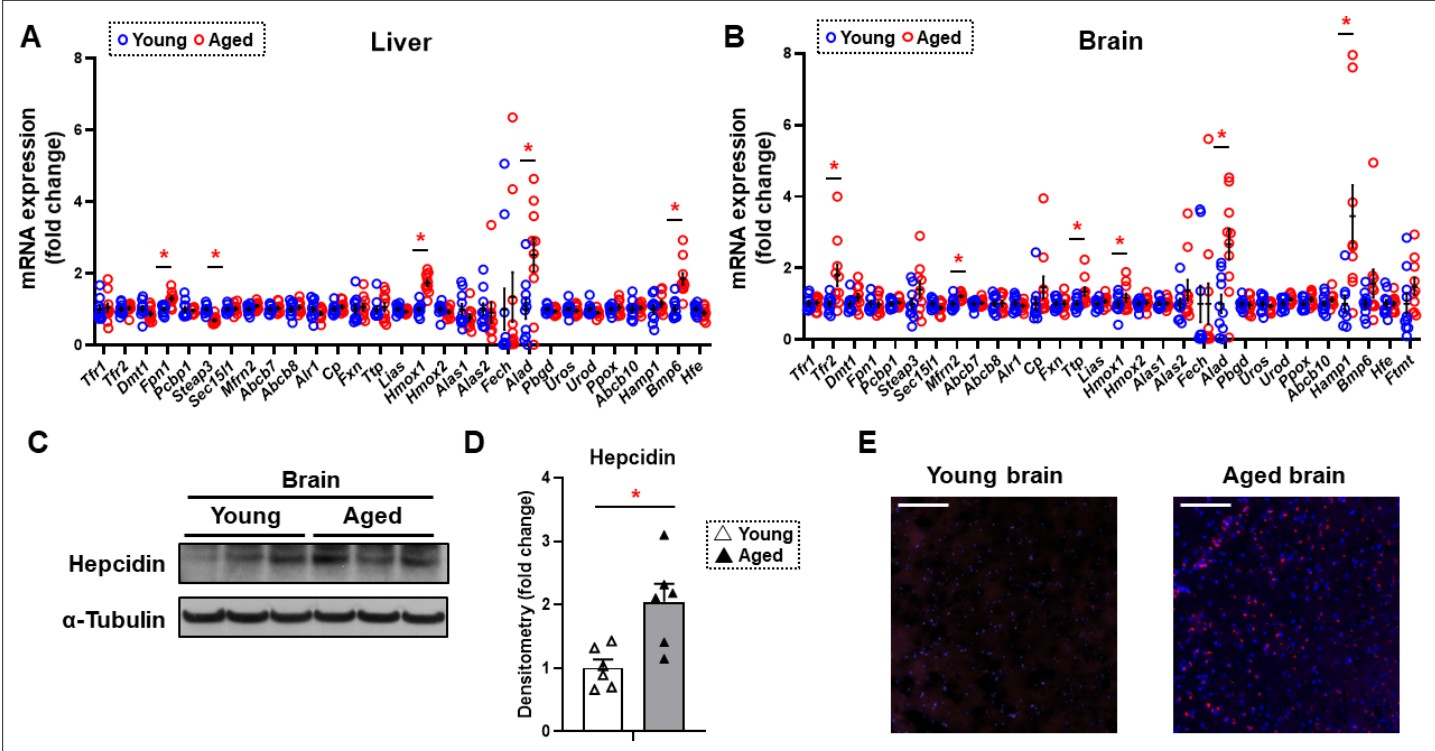

**Figure 2.** Hepcidin protein expression is significantly increased in the aged brain cortex. mRNA levels of proteins involved in iron regulation in the liver (**A**) and brain (**B**) in young (4 months old) and old (22 months old) mice (n=10/group). Means and SEM are indicated as horizontal and vertical bars, respectively. An undetected measurement (n=1) in the brain Fech and undetected measurements (n=2) and an outlier (n=1) in the brain Hamp1 were excluded. Tfr1 = transferrin receptor 1, Tfr2 = transferrin receptor 2, Dmt1 = divalent metal transporter 1, Fpn1 = ferroportin 1, Pcbp1 = Poly(RC) Binding Protein 1, Steap3 = Metalloreductase Six-Transmembrane Epithelial Antigen Of Prostate 3, Sec15l1 = exocyst complex component 6, Mfrn2 = mitoferrin 2, Abcb7 = ATP-binding cassette sub-family B member 7, Abcb8 = ATP-binding cassette sub-family B member 8, Alr1 = augmenter of liver regeneration, Cp = ceruloplasmin, Fxn = frataxin, Ttp = tristetraprolin, Lias = tristetraprolin, Hmox1 = heme oxygenase 1, Hmox2 = heme oxygenase 2, Alas1 = 5'-aminolevulinate synthase 1, Alas2 = 5'-aminolevulinate synthase 2, Fech = ferrochelatase, Alad = aminolevulinate dehydratase, Pbgd = porphobilinogen deaminase, Uros = uroporphyrinogen III synthase, Urod = uroporphyrinogen decarboxylase, Ppox = protoporphyrinogen oxidase, Abcb10 = ATP-binding cassette, sub-family B member 10, Hamp1 = hepcidin1, Bmp6 = bone morphogenetic protein 6, Hfe = homeostatic iron regulator, Ftmt = mitochondrial ferritin. (**C**) Representative immunoblot for hepcidin1 in the brain (n=6). (**D**) Summary of densitometry analysis of panel (**C**). (**E**) Representative immunohistochemistry of hepcidin1 (Red = anti-hepcidin1, blue = DAPI) in the brain frontal cortex of young and aged mice. Scale bar=200 μm. * p<0.05.

The online version of this article includes the following source data for figure 2:

**Source data 1.** Full-length images of immunoblots shown in *Figure 2*.

**Source data 2.** qRT-PCR liver – Original data of qRT-PCR in the liver shown in *Figure 2*.

**Source data 3.** qRT-PCR brain – Original data of qRT-PCR in the brain shown in *Figure 2*.

**Source data 4.** Densitometry – Original data of densitometry quantifications of immunoblots shown in *Figure 2*.

to transferrin-receptor protein (TfRC), since it has a much lower affinity for transferrin (*Kawabata et al., 1999*). Instead, it acts as a signaling molecule to regulate *Hamp1* transcription through SMAD signaling (*Silvestri et al., 2014*). Since both *Tfr2* and *Hamp1* are increased in the brain cortex with aging, we then focused our studies on the cellular hepcidin pathway. To confirm that the increase in *Hamp1* mRNA with aging is associated with an increase in its protein, we measured HAMP1 protein in the tissues of brain cortex in young and aged mice and noted a significant increase in its protein levels (*Figure 2C and D*). Finally, we performed immunohistochemistry on the brain frontal cortex of aged mice and confirmed that HAMP1 protein levels are significantly increased with aging (*Figure 2E*).

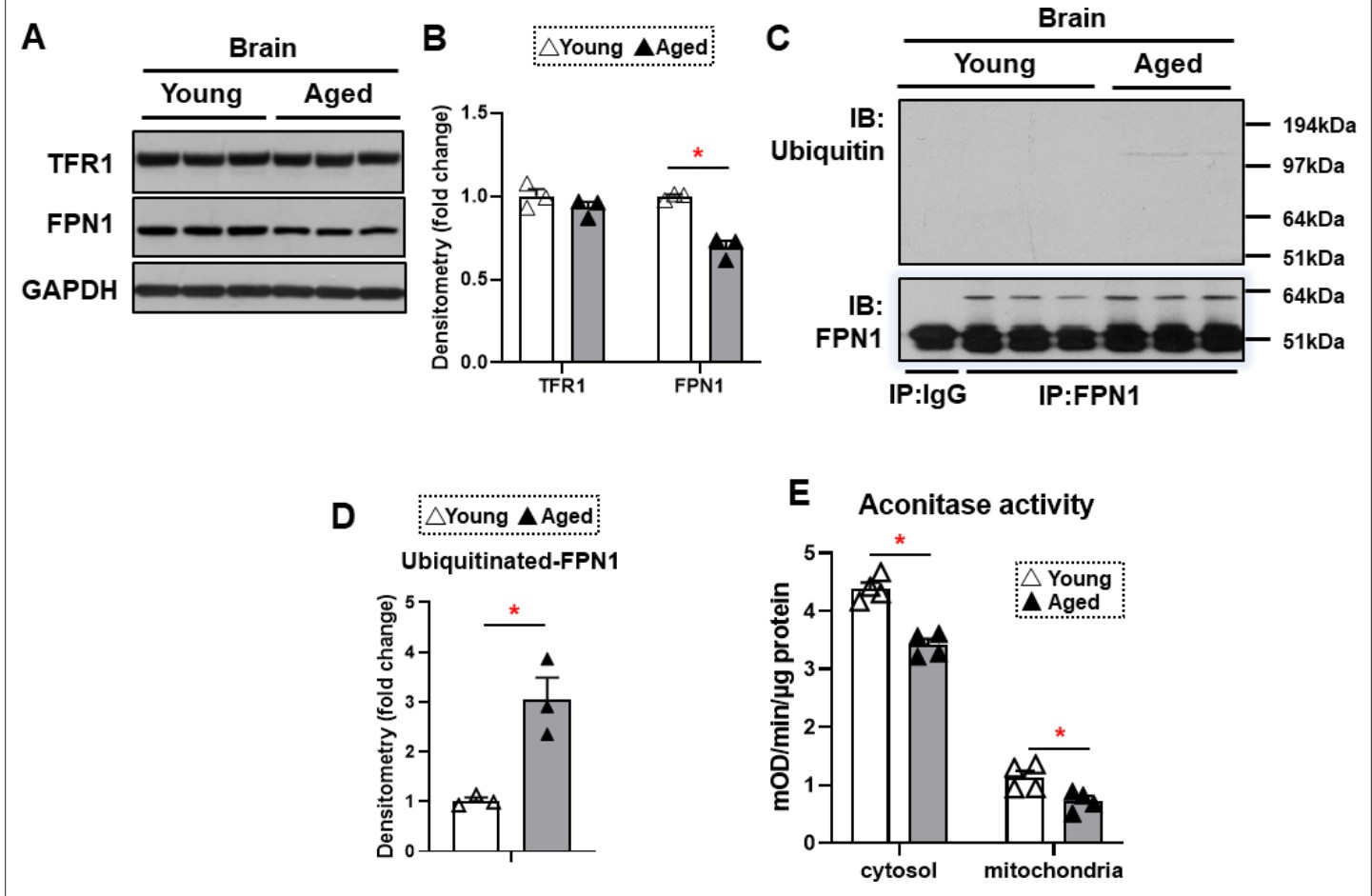

**Figure 3.** FPN1 protein level is decreased while its poly-ubiquitination is increased in the brain cortex of aged mice. (**A**) Immunoblots of iron transporting proteins TfR1 and FPN1 in the brain cortex of young and aged mice (n=3). (**B**) Summary of densitometric analysis of panel (**A**). (**C**) Poly-ubiquitination levels of FPN1, as assessed by immunoprecipitation, in the brain cortex of young and aged mice (n=3). (**D**) Summary of the densitometric analysis of panel (**C**). (**E**) Fe-S cluster containing aconitase enzyme activity, a marker of cellular oxidative stress, in the brain cortex of young and aged mice (n=4). c-aconitase = cytosolic aconitase (ACO1), m-aconitase = mitochondrial aconitase (ACO2). * p<0.05.

The online version of this article includes the following source data for figure 3:

**Source data 1.** Full length images of immunoblots shown in *Figure 3*.

**Source data 2.** Densitometry – Original data of densitometry quantifications of immunoblots shown in *Figure 3*.

**Source data 3.** Aconitase assay – Original data of aconitase enzyme activities shown in *Figure 3*.

## Aging is associated with decreased FPN1 protein levels through its ubiquitination

The major effect of hepcidin protein on iron regulation is through ubiquitination and subsequent degradation of FPN1, resulting in a decrease in cellular iron export (*Qiao et al., 2012*). This mechanism of hepcidin effect generally occurs in intestinal cells and macrophages, which leads to decreased iron export from these cells. Since HAMP1 protein levels are increased in the brain cortex of aged mice, we hypothesized that FPN1 levels would be decreased in the brains of these mice. FPN1 protein levels are significantly decreased in the brain cortex tissue of aged mice, while TfR1 levels are unchanged (*Figure 3A and B*). Since mRNA levels of FPN1 are not significantly changed in the brain cortex, we hypothesized that the decrease in FPN1 protein might be due to its degradation by hepcidin. We therefore measured FPN1 protein ubiquitination in the brains of aged mice, and showed a significant increase in its ubiquitination (*Figure 3C and D*). These results indicate that FPN1 protein is ubiquiti-nated and decreased with aging, perhaps as a consequence of the higher levels of hepcidin.

We then assessed whether the increase in iron in the brain cortex of aged mice is associated with increased oxidative stress. Aconitase is a Fe/S-containing protein that is highly sensitive to oxidative stress (*Noster et al., 2019*). Our results demonstrated a decrease in both cytosolic and mitochondrial aconitase activity in the brain cortex of aged mice (*Figure 3E*), being consistent with the notion that oxidative stress is increased in the aged brain (*Kandlur et al., 2020*).

## Discussion

Iron is critical for normal cell function, but its excess can lead to cellular damage due to oxidative stress. Thus, the levels of cellular iron, particularly in the mitochondria, are tightly regulated (*Koleini et al., 2021*). Iron has been implicated in the development of some neurodegenerative diseases that are associated with aging, including Alzheimer's and Parkinson's diseases (*Xu et al., 2012*). However, the molecular mechanism for the increase in brain iron in aging is not known. In this paper, we showed that iron accumulates in the mouse brain cortex with aging and that this is associated with a significant increase in hepcidin production in the brain cortex. Since the major function of hepcidin is to bind to FPN1, leading to FPN1 degradation, we assessed levels of FPN1 in extracts of brain cortex from young and aged mice and showed that while its mRNA is not changed, protein levels of FPN1 are significantly decreased. We also demonstrate that brain FPN1 ubiquitination is increased with aging. Finally, we demonstrate that there is increased oxidative stress in the brains of aged mice, which may well reflect, at least in part, accumulation of iron, perhaps in the context of other age-dependent changes in oxidative damage and defenses against oxidative injury.

Hepcidin antimicrobial peptide (HAMP1) is a small 25–amino acid hormone peptide that is secreted by hepatocytes in conditions of iron sufficiency, and which binds to FPN1 and promotes its internalization and degradation. FPN1 degradation inhibits iron absorption from the digestive tract and the release of iron from macrophages. Although liver is believed to be a major source of systemic hepcidin production, this peptide has recently been shown to be expressed by other tissues. A recent study demonstrated that hepcidin is expressed in the heart and that its cardiac-specific deletion has detrimental effects on cardiac function (*Lakhal-Littleton et al., 2016*). Additionally, hepcidin has been inferred to play a role in colorectal cancer by sequestering iron to maintain the nucleotide pool and sustain proliferation in colorectal cancer (*Schwartz et al., 2021*). Recent studies have also demonstrated that hepcidin overexpression in astrocytes protects against amyloid-β induced brain damage in mice and also protects against mouse models of Parkinson's disease (*Liang et al., 2020*; *Zhang et al., 2020*). On the other hand, it has been reported that upregulation of local hepcidin expression is associated with brain iron accumulation in patients with Alzheimer's disease (*Chaudhary et al., 2021*). However, it was not known whether hepcidin is increased with changes in its target FPN1 in the aged brain in mice. In the present study, our data show that brain hepcidin is increased with aging and that it likely leads to a reduction in FPN1 levels and iron accumulation. Since the results of this study are based on a small number of immunoblot samples using single-gender of mice, additional studies are needed to confirm these findings in a larger cohort of mice of both genders.

Previous reports have suggested that inflammation plays a role in the brain-derived hepcidin expression via lipopolysaccharide- or interleukin-6-mediated STAT3 pathway (*Lieblein-Boff et al., 2013*; *Wang et al., 2008*), and another study suggested that toll-like receptor 4/myD88-mediated signaling of hepcidin expression cause brain iron accumulation and oxidative injury (*Xiong et al., 2016*). Thus, age-related inflammation may be a regulator of iron accumulation in the brain via increased brain-derived hepcidin expression and subsequent FPN1 downregulation. In contrast, BMP6 secreted by the liver sinusoidal endothelial cells in response to iron binds to BMPR and induces hepcidin production via the SMAD pathway. However, it is not known whether BMP6 regulates hepcidin expression in the brain. Indeed, our data showed that gene expression of BMP6 was not increased in the aged brain. Additionally, TfR2 is associated with positive regulation of hepcidin expression (*Silvestri et al., 2014*), and mitochondrial iron transport in the brain (*Mastroberardino et al., 2009*). However, the expression of brain hepcidin is altered in TfR2-deficient mice in response to systemic iron deficiency or iron overload (*Pellegrino et al., 2016*), suggesting that TfR2 is not the sole regulator of brain hepcidin expression. Further studies are needed to clarify whether increased TfR2 in the aged brain contributes to the increase in brain-derived hepcidin expression or alternatively contributes to iron accumulation with age.

*Lu et al., 2017* have shown that, compared to younger animals, 24-month-old rats have significant increase in hepcidin expression in the brain cortex, in addition to hippocampus, striatum, and substantia nigra. However, FPN1 levels were not significantly changed in different brain regions. The authors concluded that increased DMT1, rather than TfR1 or FPN1, may be responsible for the increased brain iron with aging. Although the increase in hepcidin is consistent with our data, their proposed mechanism differs from what we observed, which could be due to differences in species, and sample size between our studies. Additionally, our data showed that mRNA and protein levels of TfR1 do not change with aging. TfR1 is regulated by both iron-dependent and iron-independent mechanisms. For iron-dependent pathways, IRPs are well-known positive regulators of TfR1. However, TfR1 is also regulated by iron-independent immune response, such as NF-kB- and/or HIF-mediated pathways (*Samavati et al., 2008*). Since we found that decreased IRP2 in the aged brain, it is possible that TfR1 gets downregulated via decreased IRP2-mediated pathway, but iron-independent regulation of TfR1 cancels IRP2-mediated decrease in the aged brain. Our data therefore cannot resolve whether TfR1 is central to age-mediated iron accumulation in the brain and if so, by what mechanism. Additional studies are necessary to clarify this important point.

A recent study reported that in patients with Parkinson's disease, the activity of heme oxidase-1, which is regulated by HMOX1 and degrades heme, is increased in the serum and brain and its activation is associated with anemia and iron accumulation in the substantia nigra, but not other brain regions (*Xu et al., 2021*). Considering that cytosolic and mitochondrial heme iron levels were not changed with aging in both liver and brain in the present study, the increase in ALAD expression (which can catalyze heme synthesis) might be a compensatory mechanism against increased degradation of heme via increased HMOX1. However, the changes noted in HMOX1 and ALAD would not explain the increase in non-heme iron in the brain.

The mechanism for iron-mediated aging is likely due to the oxidative effects of iron and cellular damage. Mitochondria are the major site of reactive oxygen species production in the cells, thus, the levels of mitochondrial iron are tightly controlled. In our studies, we noted an increase only in the cytosolic non-heme iron in the liver and the skeletal muscle of aged mice, while non-heme iron was increased in both the mitochondrial and cytosolic fraction of the brain cortex tissue. The significance of sole increase in cytosolic iron is not clear, but the increase in mitochondrial iron in the brain is significant since it likely contributes to the oxidative damage that occurs in this tissue with aging. It is also possible that cytosolic iron accumulation in the liver and muscle tissue is due to mild increased systemic iron. Since the brain is not an iron storage organ and iron movement to the brain has to go through the blood-brain barrier (BBB), iron accumulation in the brain with aging is likely independent of systemic iron metabolism.

Our studies have clinical implications for neurodegenerative diseases associated with aging through two possible mechanisms: (1) by targeting brain iron with iron chelators that can cross the BBB, and (2) through a reduction in locally produced hepcidin. There is currently interest in taking the first approach and clinical studies support this approach (*Sun et al., 2018*). However, targeting hepcidin in the brain is technically challenging and may require the design of small molecule hepcidin inhibitors that can also cross the BBB. We also acknowledge that we do not identify the specific cells in the brain that express hepcidin and FPN1. Further studies are needed to address cellular specificity and interaction among increased brain-derived hepcidin regulation, *Fpn1* expression, and cytosolic and mitochondrial iron accumulation with aging. Identification of these pathways may have therapeutic potentials.

## Materials and methods

**Key resources table**

| Reagent type (species) or resource | Designation | Source or reference | Identifiers | Additional information |
|---|---|---|---|---|
| Strain, strain background (*Mus musculus*, female, C57BL/6) | UM-HET3 | Dr. Miller lab | | *Harrison et al., 2021* (cited in the paper) |

*Continued on next page*

*Continued*

| Reagent type (species) or resource | Designation | Source or reference | Identifiers | Additional information |
|---|---|---|---|---|
| Strain, strain background (*M. musculus*, female, C57BL/6) | Wild type | Jackson Laboratories | | |
| Antibody | Rabbit polyclonal anti-HPRT antibody | ProteinTech | 150-59-1-AP | WB (1:5000) |
| Antibody | Rabbit polyclonal anti-SDHA antibody | Invitrogen | 45-920-0 | WB (1:1000) |
| Antibody | Rabbit polyclonal anti-FTN antibody | Sigma-Ardrich | F5012 | WB (1:1000) |
| Antibody | Rabbit polyclonal anti-IRP2 antibody | Sigma-Ardrich | SAB2101174 | WB (1:500) |
| Antibody | Rabbit polyclonal anti-TFR1 antibody | ProteinTech | 100-84-2-AP | WB (1:1000) |
| Antibody | Rabbit polyclonal anti-FPN1 antibody | Novus Biologicals | NBP1-21502 | WB (1:1000) |
| Antibody | Rabbit monoclonal anti-Hepcidin antibody | Abcam | 190775 | WB (1:200) IH (1:20) |
| Antibody | Mouse polyclonal anti-ubiquitin antibody | Cell Signaling Technologies | 3933S | WB (1:1000) |
| Antibody | Mouse monoclonal anti-GAPDH antibody | ProteinTech | 60004-1-Ig | WB (1:5000) |
| Antibody | Rabbit polyclonal anti-α-Tubulin antibody | ProteinTech | 66031-1-Ig | WB (1:5000) |
| Antibody | HRP-conjugated donkey polyclonal anti-rabbit IgG antibody | Jackson ImmunoResearch | 711-035-152 | WB (1:5000) |
| Antibody | HRP-conjugated donkey polyclonal anti-mouse IgG antibody | Jackson ImmunoResearch | 715-035-150 | WB (1:5000) |
| Antibody | Alexa Fluor 594 goat polyclonal anti-rabbit IgG | Jackson ImmunoResearch | 111-585-144 | IH (1:200) |
| Chemical compound, drug | 3-(2-Pyridyl)–5,6-di(2-furyl)–1,2,4-triazine-5′,5″-disulfonic acid disodium salt (Ferrozine) | Sigma-Aldrich | 82940 | |
| Chemical compound, drug | Trichloroacetic acid | Sigma-Aldrich | T6399 | |
| Chemical compound, drug | Thioglycolic acid | Sigma-Aldrich | T3758 | |
| Chemical compound, drug | Hemin | Sigma-Aldrich | H9039 | |
| Chemical compound, drug | RNA-STAT60 | Teltest | Cs-502 | |

*Continued on next page*

*Continued*

| Reagent type (species) or resource | Designation | Source or reference | Identifiers | Additional information |
|---|---|---|---|---|
| Chemical compound, drug | Glycogen | Life Technologies | AM9510 | |
| Chemical compound, drug | Paraformaldehyde | Thermo Fisher Scientific | AC416780250 | |
| Chemical compound, drug | RIPA Buffer | Thermo Fisher Scientific | 89901 | |
| Commercial assay or kit | ProteaseArrest Protease Inhibitor | G-Biosciences | 786-437 | |
| Commercial assay or kit | qScript cDNA Synthesis Kit | Quanta | 95047-500 | |
| Commercial assay or kit | PerfeCTa SYBR Green FastMix | Quanta | 95074-05K | |
| Commercial assay or kit | SuperSignal West Pico PLUS Chemiluminescent Substrate | Pierce | 34579 | |
| Commercial assay or kit | Dynabeads Protein G for Immunoprecipitation | Invitrogen | 10003D | |
| Commercial assay or kit | BCA Protein Assay Kit | Pierce | 23225 | |
| Commercial assay or kit | Mitochondria Isolation Kit for Tissue | Pierce | 89801 | |
| Commercial assay or kit | Aconitase Activity Assay Kit | Abcam | Ab109712 | |
| Commercial assay or kit | OTC compound | Sakura Finetek | 4583 | |
| Sequence-based reagent | Primers for qRT-PCR | This manuscript | N/A | Included in the next table |
| Software, algorithm | GraphPad Prism | GraphPad | Version 9 | |
| Software, algorithm | ImageJ | NIH | 1.53c | |

## Primer sequences

| Genes | Forward primer | Reverse primer |
|---|---|---|
| *Actb* | CCGTGAAAAGATGACCCAGAT | GTACATGGCTGGGGTGTTG |
| *Hprt1* | CTGGAAAGAATGTCTTGATTGTTG | TGCATTGTTTTACCAGTGTCAA |
| *Tfr1* | GCATTGCGGACTGTAGAGG | GCTTGATCCATCATTCTCAGC |
| *Tfr2* | AGCCCATCAGTGCTGACATT | AGGAGAGCCTGAGAGGTGAC |
| *Dmt1* | GGCGTGTGTGGAGGTGGCGG | TGGTCCCCAGAAGCGCCATCG |
| *Fpn1* | TGGCCACTCTCTCTCCACTT | ACACTGCAAAGTGCCACATC |
| *Pcbp1* | AGATCAAAATTGCCAACCCG | AGCCAGTAATAGTGACCTGC |
| *Steap3* | AGGAGTTCAGCTTCGTGCAG | GAGGGCTAGACAAGATGCGTA |
| *Sec15l1* | CCACCCTCCGATCTGTGTAT | AAACTTCTTGTGTGCGTTTGG |
| *Mfrn2* | AGTGACGTAATCCACCCAGGGGC | CTCTGCTTGACGACTTCCGCTGG |

*Continued on next page*

*Continued*

| Genes | Forward primer | Reverse primer |
| --- | --- | --- |
| Abcb7 | GGAGGAGGACTCCACACAGA | ATGGCAACTCTGGCTCGTAG |
| Abcb8 | GGGCAACAGGTGTAGCAGAT | TGCTTGATAGCGTTCCTCCT |
| Alr1 | CCCTGCGAGGAATGTGCGGAA | TCACCTCATTGTGCAGGCGGC |
| Cp | TGCTCCTTCTGGGACGGACATCTTC | GCCACCAATTCTTGTGGCACCTTGC |
| Fxn | CAGACAAGCCCTATACCCTG | AGCCAGATTTGCTTGTTTGG |
| Ttp | CCATCTACGAGAGCCTCCAG | CGTGGTCGGATGACAGGT |
| Lias | AGGAAACTTAAAGCGCCAGA | GCCATGGAGGTAGTCTTAGCC |
| Hmox1 | AGGCTTTAAGCTGGTGATGG | CTTCCAGGGCCGTGTAGATA |
| Hmox2 | TGGCACCAGAAAAGGAAAAC | CTTCCTTGGTCCCTTCCTTC |
| Alas1 | TGGGGCCAAGCCAGCTCCTC | AGGAGCCTGCTGGACTGCGG |
| Alas2 | TTGGTTCGTCCTCAGTGCAGGG | GCCACCATCCTGAGCCCAAAGTC |
| Fech | CCGGAAATGCTTTCGGCCAGCG | ACAGGCCCTTGAGCTGCCGTG |
| Alad | CGCTTTTAGAGCGGGAGAGC | AGAGAAGTCTGCTGGATGGAG |
| Pbgd | AGCTACAGAGAAAGTTCCCC | ACTGAATTCCTGCAGCTCAT |
| Uros | GAGGACTCATTTTCACCAGC | GACTTGGCATTCCATCTGTC |
| Urod | TTACTGTTTACCATGGAGGC | AGGAACGTGTCATTCTTCAG |
| Ppox | ACCTAGCAAGTAAAGGGGTC | CTTATAATGTGGTCAGCCTCC |
| Abcb10 | ACCGGTGTGCGAGACCTTGGG | TGGACACAGCCAGAAACCCAACTGC |
| Hamp1 | TTTGCACGGGGAAGAAAGCA | GTGGCTCTAGGCTATGTTTTGC |
| Bmp6 | GCTTTGTGAACCTGGTGGAG | GTCGTTGATGTGGGGGAGAAC |
| Hfe | CCTCCACGTTTCCAGATCCT | CTCTGAGGCACCCATGAAGAG |
| Ftmt | ATTTCCCCAGCCTGAAAGAT | TCCCATCCCTGTCATACCAT |

## Tissues from young and aged mice

Tissues of liver, gastrocnemius muscle, and dissected brain cortex from young (4 months old) and aged (22 months old) female UM-HET3 mice, which is a genetically heterogeneous mouse model that is the first-generation offspring of a CByB6F1 × C3D2F1 cross to produce a diverse heterogeneous population in the Interventions Testing Program (ITP; https://www.nia.nih.gov/research/dab/interventions-testing-program-itp) (*Harrison et al., 2021*), were used in the present study. Mice were given diet Purina 5LG6 containing 345 ppm iron, which is similar to the published normal iron diet (*Gutschow et al., 2015*). For hepcidin immunohistological analysis, young (4 months old) and aged (22 months old) female C57BL/6 mice were used. This study was performed in strict accordance with the recommendations in the Guide for the Care and Use of Laboratory Animals of the National Institutes of Health. Our research was approved by the University of Michigan's Institutional Animal Care and Use Committee. The approval number currently associated with this activity is PRO00009981.

## Cytosolic and mitochondrial fractioning

Cytosolic and Mitochondrial fractions from tissues were isolated using Mitochondrial Isolation Kit for Tissues (Pierce) according to the manufacturer's protocol.

## Non-heme and heme iron measurement

Iron contents in the cytosolic and mitochondrial fractions of the tissues were measured as previously described (*Bayeva et al., 2012*; *Sato et al., 2018*). Briefly, for the non-heme iron measurements, equal amounts of protein were mixed with protein precipitation solution (1:1 1 N HCl and 10% trichloroacetic acid) and heated at 95°C for 1 hr to release iron. The precipitated proteins were removed by

centrifugation at 16,000×*g* for 10 min at 4°C, the supernatant was mixed with the equal volume of chromogenic solution (0.5 mM ferrozine, 1.5 M sodium acetate, 0.1% [v/v] thioglycolic acid), and the absorbance was measured at 562 nm. For the heme iron measurements, equal amounts of protein were mixed with 2.0 M oxalic acid and were heated at 95°C for 1 hr to release iron from heme and generate protoporphyrin IX. Samples were then centrifuged at 1000×*g* for 10 minat 4°C and the fluorescence of the supernatant was assessed at 405 nm/600 nm.

## Reverse transcription and quantitative real-time PCR

RNA was isolated from tissues using RNA-STAT60 (Teltest). Reverse transcribed with qScript Reverse Transcription Kit (Quanta) according to the manufacturers' instruction, and as described previously (*Chang et al., 2021*). The resulting cDNA was amplified quantitatively using PerfeCTa SYBR Green Mix (Quanta) on a 7500 Fast Real-Time PCR System (Applied Biosystems). The relative gene expression was determined using differences in Ct values between gene of interest and housekeeping control genes β-actin and hypoxanthine phosphoribosyltransferase-1 (*Hprt1*).

## Immunoblotting

Tissues were lysed in radioimmunoprecipitation assay (RIPA) buffer supplemented with protease inhibitor (Thermo Fisher Scientific). Protein concentration in lysates was determined using BCA Protein Quantification Kit (Pierce) and as described previously (*Sawicki et al., 2018*). Equal amounts of proteins were resolved on 4–12% Novex Bis-Tris poly-acrylamide gel (Invitrogen) and blotted onto nitrocellulose membrane (Invitrogen). After blocking with tris-buffered saline containing 0.05% Tween 20 (Thermo Fisher Scientific) and 5% milk, the membrane was incubated overnight at 4°C in primary antibody against HPRT (ProteinTech), succinate dehydrogenase complex flavoprotein subunit A (SDHA, Invitrogen), FTN (Sigma-Aldrich), IRP2 (Novus Biologicals), TfR1 (Invitrogen), FPN1 (Novus Biologicals), Hepcidin (Abcam), Ubiquitin (Cell Signaling Technology), GAPDH (ProteinTech), and α-Tubulin (ProteinTech). The following day, membranes were incubated with HRP-conjugated anti-mouse or anti-rabbit secondary antibodies (Jackson ImmunoResearch) for 1 hr and proteins were visualized using Super Surgical Western Pico ECL substrate (Pierce). Quantification of immunoblotting image was done using ImageJ (NIH).

## Immunoprecipitation of FPN1

Tissues were lysed in IP buffer containing 25 mM Tris-HCl (pH 7.5), 150 mM NaCl, 1 mM EDTA, 0.1% NP-40 with protease inhibitor cocktail (G-Biosciences). 400 μg of protein was pre-incubated with 40 μl of protein G magnetic beads (Invitrogen) for 1 hr at 4°C. After the beads had been discarded, the supernatant was incubated with FPN antibody (Novus Biologicals) or IgG at a rotator overnight at 4°C. The mixture was then incubated with 40 μl of fresh beads for 1 hr at 4°C. A magnetic field was applied to this mixture, and the supernatant was removed. The magnetic beads were washed three times with 400 μl of IP buffer, re-suspended in 40 μl of SDS sample buffer, and incubated for 10 min at 70°C. Finally, 20 μl of the supernatant was collected after applying a magnetic field to the mixture and was used for immunoblotting.

## Immunohistochemistry of hepcidin

Mice were anesthetized with an intraperitoneal injection of 250 mg/kg dose of freshly prepared Tribromoethanol (Avertin). They were then transcardially perfused with ice-cold phosphate-buffered saline (PBS) to wash out the blood, and following the discoloration of liver, the buffer was replaced to freshly prepared ice-cold 4% paraformaldehyde (PFA) with the assistance of circulating pump to supply sufficient perfusion pressure. After 20 min perfusion, the brain tissue was extracted from the skull and was fixed overnight in 4% PFA. The fixed brain tissue was placed in 30% sucrose for 48 hr at 4°C. Brain tissue was then submerged into OTC compound (Sakura Finetek, Torrance, CA) and frozen in liquid nitrogen. 30 μm coronal sections were cut using a cryotome (Leica) and mounted on Fisherbrand Superfrost Plus Microscope Slides. Sections were permeabilized using 0.25% Triton-x100 in PBS for 30 min, washed three times in PBS, and incubated with primary antibody against hepcidin overnight at 4°C. Sections were then washed three times in PBS and incubated with Alexa Fluor594 Goat Anti-Rabbit IgG at 1:200 (Jackson ImmunoResearch) for 2 hr at room temperature. Nuclei were counterstained with DAPI containing ProLong Gold Antifade mounting media (Invitrogen) and images of frontal cortex were acquired using a Zeiss Axio Observer.Z1 fluorescence microscope.

## Aconitase activity measurements

Aconitase activities in cytosolic and mitochondrial fraction of the brain cortex lysates were measured using Aconitase Enzyme Activity Microplate Assay Kit (Abcam) according to the manufacturer's instructions.

## Statistical analysis

Data are presented as mean ± SEM. There was no randomization of the samples, but the investigator performing the measurements was blinded to the group assignment. For sample size estimation, our previous study (*Chang et al., 2016*) indicated that changes in mitochondrial non-heme iron contents of about 20% can affect iron-dependent cytotoxicity in the mouse heart. Given an estimated standard deviation of iron of 15% in each group, we estimated that six samples would be needed to detect a 20% difference in change of iron contents. For other experiments, power analysis was not performed a priori because we could use sufficient numbers of stored tissue samples that had already been cryo-preserved and there was no risk of wasting unnecessary animals. We determined n=10 for the RNA levels and n=3–6 for the immunoblots and enzyme activities for our analysis. This was based on our prior experience performing similar experiments. Unpaired two-tailed Student's t-tests were used to determine statistical significance. $p < 0.05$ was considered to be statistically significant, as indicated by an asterisk. Analysis was performed using Graphpad Prism 9.

## Acknowledgements

The authors would like to thank Chunlei Chen for technical help. TS was supported by American Heart Association. HA is supported by NIH R01 HL127646, R01 HL140973, and R01 HL138982, and a grant from Leducq foundation. RAM was supported by NIH grants U01-AG022303-17 and P30-AG024824.

## Additional information

### Competing interests

Hossein Ardehali: Reviewing editor, *eLife*. The other authors declare that no competing interests exist.

### Funding

| Funder | Grant reference number | Author |
|---|---|---|
| NIH Office of the Director | NIH R01 HL127646 | Hossein Ardehali |
| Leducq Foundation | | Hossein Ardehali |
| NIH Office of the Director | R01 HL140973 | Hossein Ardehali |
| NIH Office of the Director | R01 HL138982 | Hossein Ardehali |

The funders had no role in study design, data collection and interpretation, or the decision to submit the work for publication.

### Author contributions

Tatsuya Sato, Data curation, Formal analysis, Investigation, Methodology, Validation, Visualization, Writing - review and editing; Jason Solomon Shapiro, Conceptualization, Data curation, Formal analysis, Investigation, Methodology, Validation, Visualization, Writing - review and editing; Hsiang-Chun Chang, Conceptualization, Methodology, Validation, Writing - review and editing; Richard A Miller, Formal analysis, Resources, Writing - review and editing; Hossein Ardehali, Conceptualization, Formal analysis, Funding acquisition, Investigation, Project administration, Supervision, Validation, Writing - original draft, Writing - review and editing

### Author ORCIDs

Tatsuya Sato ⓘ http://orcid.org/0000-0001-7876-1772
Jason Solomon Shapiro ⓘ http://orcid.org/0000-0003-0880-3142
Hsiang-Chun Chang ⓘ http://orcid.org/0000-0002-9201-4500
Hossein Ardehali ⓘ http://orcid.org/0000-0002-7662-0551

## Ethics

This study was performed in strict accordance with the recommendations in the Guide for the Care and Use of Laboratory Animals of the National Institutes of Health. All of the animals were handled according to approved institutional animal care and use committee (IACUC) of Northwestern University. PRO00009981.

## Decision letter and Author response

Decision letter https://doi.org/10.7554/eLife.73456.sa1
Author response https://doi.org/10.7554/eLife.73456.sa2

---

# Additional files

## Supplementary files

• Transparent reporting form

## Data availability

All data generated or analysed during this study were included as figures in the paper. All of the original datasets are included in the manuscript and supporting files.

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
