## [Editor Report]

The authors present a manuscript addressing an important unmet need, aiming to understand the mechanism of neurodegenerative disease, specifically focused on the role of iron overload in the aging brain, that is of interest to basic iron biologists and neurologists as a Short Report of novel significance. The current study demonstrates increased cytosolic and mitochondrial non-heme iron only in the aging brain, increased cortical hepcidin expression, and decreased levels of FPN1, together supporting a hypothesis that cortical hepcidin sequesters iron in neuronal cells and is associated with aging.

---

## [Decision Letter]

**Decision letter after peer review:**

Thank you for submitting your article "Aging is associated with increased brain iron through brain- derived hepcidin expression" for consideration by *eLife*. Your article has been reviewed by 3 peer reviewers, one of whom is a member of our Board of Reviewing Editors, and the evaluation has been overseen by Mone Zaidi as the Senior Editor. The following individual involved in review of your submission has agreed to reveal their identity: Joshua L Dunaief (Reviewer #2).

Essential revisions:

The reviewers found significant merit in the work. However, the evaluation identified sufficient weaknesses that prevent publication of this work in the present form. The main shortcomings of the presented work include the relatively underdeveloped cell-specificity of hepcidin expression in the brain and lack of mechanistic information explaining hepcidin upregulation. In this Short Report format, it would be reasonable for the authors to extend the discussion regarding the shortcomings of the current approach, especially addressing the prior art and aligning a more modest conclusion given the relatively superficial presentation. In addition, it would be essential that the revision included: (1) delineating which part of the brain was dissected for these studies, (2) validation of the purity / enrichment of the mitochondrial fraction, (3) addition of means or changing data to δ δ Ct in place of "mRNA expression" in Figure 2A and B graphs, (4) discussion of why TFR1 is not decreased in the aged iron overloaded mice (Figure 3A), and (5) including additional corroborating markers of oxidative stress (in addition to aconitase activity) would increase enthusiasm. A complete list of reviewer comments can be found below.

*Reviewer #1 (Recommendations for the authors):*

Why is it important that cytosolic heme iron in the muscle is not found? Can the authors clarify and add to the Discussion section?

Decreased Steap3, an endosomal ferrireductase, is associated with anaemia due to decreased import of ion into erythroblasts (Blanc AJH 2015). This is worth at least a mention in the discussion.

The authors state that the same genes in the liver are not related to iron loading as the genes in the brain that are (e.g. Hox1 and Alad). Please edit.

As changes in TFR2 are possible evidence of a separate topic, this reviewer would recommend perhaps changing the language in this part of the Results section to state that TFR2 is being fully explored in a separate manuscript or the like.

*Reviewer #2 (Recommendations for the authors):*

1) Cite a journal article instead of: (https://www.jax.org/news-and-insights/jaxblog/2017/november/when-are-mice-considered-old#).

2) The following statement is necessarily correct: "These results suggest that the more oxidative form of iron (i.e., non-heme iron) is increased in the brain cytosol and mitochondria with aging." Heme can also be an oxidant.

3) In one place, ferroportin is misspelled as "feroportin"

4) Means should be shown on the graph for Figure 2 A and B

*Reviewer #3 (Recommendations for the authors):*

The manuscript by Sato et al. addresses an important question related to brain iron accumulation with aging. This is a timely question because iron accumulates in the brains of aged individuals and is believed to contribute to neurodegenerative diseases such as AD and PD.

However, the data are not clear, and do not support the major conclusion that local hepcidin is responsible for iron accumulation in the brain. Some examples are given below:

1. In Figure 1, the ferrozine method used to quantitate iron levels also measures iron released from ferritin. Iron sequestered in ferritin is in the ferric form, and not redox-active. Levels of ferritin in the brain increase with ageing, which is well-known, and could contribute to the observed increase in iron with aging noted by the authors.

2. The mitochondrial fraction was not tested for purity. Mitochondrial markers should be evaluated and compared with the cytosolic fraction to determine the efficacy of the procedure.

3. In Figure 2 A and B, the fold change in mRNA expression is not as accurate as qPCR analysis. In the methods section the authors mention that qPCR was used. In that case, the results should be presented as ΔΔCt instead of mRNA expression. Moreover, the change in mRNA expression of HAMP1 is marginal given the standard deviation.

4. In Figure 2 C, the hepcidin band is not clear. In the raw data for Figure 2, it seems that recombinant hepcidin peptide was fractionated in parallel in the last lane, though it is not labeled. It is unclear why the bands marked in the last lane of the 'middle exposure' are absent in the overnight longer exposure.

5. The brain homogenate should be depleted of capillary endothelial cells that synthesize hepcidin, or appropriate controls need to be used to remove their contribution.

6. In the immunostained brain sample in Figure 2 E, a positive control, perhaps a section of the liver from iron depleted and iron overloaded mouse would provide confidence that the reaction is from hepcidin, though it could be hepcidin from the peripheral circulation.

7. In Figure 3 A, TfR1 is expected to decrease if the cells are iron loaded. The authors need to discuss this discrepancy.

In conclusion, though the study is well conceived and asks an important question, the experimental design lacks appropriate controls, and the results do not support the conclusions drawn by the authors.

[Editors' note: further revisions were suggested prior to acceptance, as described below.]

Thank you for submitting your article "Aging is associated with increased brain iron through brain- derived hepcidin expression" for consideration by *eLife*. Your article has been reviewed by 3 peer reviewers, and the evaluation has been overseen by a Reviewing Editor and Mone Zaidi as the Senior Editor. The following individuals involved in review of your submission have agreed to reveal their identity: Joshua L Dunaief (Reviewer #2).

Essential revisions:

The reviewers appreciate the much improved revision of this manuscript. Several additional components need to be addressed.

1) Findings of significantly decreased FPN in the western from this outbred strain with N=3 biological replicates need to be repeated and confirmed. In lieu of a repeated experiment, the authors may choose to add to the Discussion section on this point that "additional experiments are needed to confirm these findings in a larger cohort of mice of both genders."

2) Given the authors' response regarding the selection of the frontal cortex as the main site of analysis, the Introduction is no longer relevant and should be edited to fall in line with the current focus, not focused on the substantia nigra and striatum.

*Reviewer #1 (Recommendations for the authors):*

The authors have substantively responded to most of the reviewers' comments. Several elements remain to be resolved.

1) The authors suggest in their response that they focused on the frontal cortex due to prior evidence of the importance of cortical atrophy with age. However, the introduction has a very different read and in fact states that the "substantia nigra and stratum" (misspelled striatum?), subcortical structures, are central to disease pathophysiology. Please edit the introduction more in line with the selected area of the brain studied in the current manuscript and to remove the substantive delineation of Alzheimer's and Parkinson's diseases which are NOT the focus of the current manuscript.

2) Change the title to "Aging is associated with increased brain iron through cortex-derived hepcidin expression."

3) Please spell out HPRT1 and SDCF in the methods section.

4) Discussion section regarding TFR1 regulation: please change "HIF-medicated" to "HIF-mediated".

5) Please add the following sentence to the Discussion section regarding TFR1, at the end: "Our data therefore cannot resolve whether TFR1 is central to age-mediated iron accumulation in the brain and if so, by what mechanism. Additional studies are necessary to clarify this important point." Alternatively, please measure other pathways that regulate TFR1.

6) Aconitase is involved in iron responsiveness regulation in other tissues (Bullock Blood 2010). As a consequence, using aconitase as a read out of oxidative stress when iron is altered between the 2 groups is insufficient. Please remove the phrase "indicating increased oxidative stress" in the last sentence of the Results section.

7) Please add "Further studies… increased TFR2 in the aged brain contributes to increased brain-derived hepcidin expression or alternatively contributes to iron accumulation with age." to the TFR2 part of the Discussion section.

8) Question about muscle was intended to understand how data in muscle is relevant in a paper about the brain. This remains unclear.

9) Please add components of mouse strain and gender and haematologica reference for iron diet to the methods section.

*Reviewer #2 (Recommendations for the authors):*

The authors have addressed a number of concerns, including identifying the region of brain studied and validating the enriched mitochondrial fraction. Identifying the cell types with increased Hamp1/Hepc mRNA and protein and decreased Fpn protein is important, but this has not been done. In addition, the important finding of decreased Fpn is shown in a Western with an N of only 3. This is inadequate, especially with an outbred strain and documented high variability in Hamp1/Hepc levels in Figure 2. The N for the Fpn Western should be increased, and should be correlated with Hamp1/Hepc levels.

---

## [Author Response]

Essential revisions:The reviewers found significant merit in the work. However, the evaluation identified sufficient weaknesses that prevent publication of this work in the present form. The main shortcomings of the presented work include the relatively underdeveloped cell-specificity of hepcidin expression in the brain and lack of mechanistic information explaining hepcidin upregulation. In this Short Report format, it would be reasonable for the authors to extend the discussion regarding the shortcomings of the current approach, especially addressing the prior art and aligning a more modest conclusion given the relatively superficial presentation. In addition, it would be essential that the revision included:1) Delineating which part of the brain was dissected for these studies,

Thank you for this comment. In the present study, we used frozen tissue from brain cortex for iron measurement and frontal cortex for immunohistochemistry of hepcidin. The purpose of the present study was to explore if and how cellular iron levels change in the brain with aging, but not in the brain of neurodegenerative diseases such as Alzheimer's or Parkinson's. Since it has been reported that age-related brain atrophy (which is associated with age-related cognitive decline) is evident in the cortex (specifically in the frontal cortex) and not in other parts of the brain (Proc Natl Acad Sci U S A. 2010 Dec 28;107(52):22682-6), we focused our studies on the frontal cortex. Our studies revealed that increased hepcidin is associated with iron-accumulation in the aged brain cortex. The information about the part of the brain is now added to the manuscript.

2) Validation of the purity / enrichment of the mitochondrial fraction,

We appreciate the careful review of our data. We have performed immunoblotting experiments using cytosolic marker (HPRT1 = hypoxanthine phosphoribosyltransferase-1) and mitochondrial marker (Succinate Dehydrogenase Complex Flavoprotein Subunit A) in the samples used for iron measurement, and confirmed the purity of mitochondrial isolation as *shown in the revised Figure 1G*.

3) Addition of means or changing data to δ δ Ct in place of "mRNA expression" in Figure 2A and B graphs,

Thank you for this comment. We have added the “means and SEM” in the revised Figure 2A and 2B.

4) Discussion of why TFR1 is not decreased in the aged iron overloaded mice (Figure 3A).

It has been known that TfR1 is regulated by both iron-dependent and iron-independent mechanisms. For iron-dependent pathways, IRPs are well-known positive regulators of TfR1. Since our data showed that the protein level of IRP2 was decreased in the aged brain, it is plausible that TfR1 would be concomitantly downregulated via decreased IRP2 activity. However, TfR1 has also been known to be regulated by iron-independent immune response, such as NF-κB- and/or HIF-medicated pathways (*J Biol Chem. 2008 Jul 25;283(30):20674-86*). Thus, it is likely that activated iron-independent regulation of TfR1 cancels IRP2-medicated regulation of TfR1, resulting in unchanged TfR1 expression in the aged brain. This discussion is now added to the revised manuscript.

5) Including additional corroborating markers of oxidative stress (in addition to aconitase activity) would increase enthusiasm. A complete list of reviewer comments can be found below.

We appreciate this suggestion. It has already been established that oxidative stress is increased in the aged brain, as summarized in the following review (*Front Mol Neurosci. 2020 Mar 18;13:41*). In addition, our data already suggest the presence of oxidative stress in the aged brain by measuring both cytoplasmic and mitochondrial aconitase activities. Therefore, we did not measure canonical oxidative stress markers such as lipid-peroxidation in the present study, and instead, cite previous reports that show age-related oxidative stress, in addition to the inclusion of aconitase activity.

Reviewer #1 (Recommendations for the authors):Why is it important that cytosolic heme iron in the muscle is not found? Can the authors clarify and add to the Discussion section?

The precise reason why we were not able to detect heme iron in the mitochondria of gastrocnemius muscle is unclear, but it could possibly due to technical limitations associated with the isolation of intact mitochondria from skeletal muscle tissue. Since gastrocnemius muscle is a fast-twitch muscle containing low concentration of mitochondria, it is also possible that low mitochondrial yield was insufficient for detecting measurable iron, especially heme iron.

Decreased Steap3, an endosomal ferrireductase, is associated with anaemia due to decreased import of ion into erythroblasts (Blanc AJH 2015). This is worth at least a mention in the discussion.

Thank you very much for this suggestion. We hope that the Reviewer agrees that since our paper focuses on brain iron and Steap3 is not changed in the brain with aging, we do not think a discussion on this protein would be relevant to the paper.

The authors state that the same genes in the liver are not related to iron loading as the genes in the brain that are (e.g. Hox1 and Alad). Please edit.

In the revised manuscript, we discuss the possible roles of HOX1 and ALAD in iron and aging.

As changes in TFR2 are possible evidence of a separate topic, this reviewer would recommend perhaps changing the language in this part of the Results section to state that TFR2 is being fully explored in a separate manuscript or the like.

Thank you for this great suggestion. In the revised manuscript, we have mentioned possible roles of TfR2 in iron accumulation in aged brain citing previous papers.

Reviewer #2 (Recommendations for the authors):1) Cite a journal article instead of: (https://www.jax.org/news-and-insights/jaxblog/2017/november/when-are-mice-considered-old#).

We thank the reviewer for this suggestion. In the revised manuscript, we cited original textbook from “Flurkey K, Currer JM, Harrison DE. 2007. The Mouse in Aging Research. In The Mouse in Biomedical Research 2nd Edition. Fox JG, et al., editors. American College Laboratory Animal Medicine (Elsevier), Burlington, MA. pp. 637–672.”, instead of citing URL.

2) The following statement is necessarily correct: "These results suggest that the more oxidative form of iron (i.e., non-heme iron) is increased in the brain cytosol and mitochondria with aging." Heme can also be an oxidant.

We agree with the Reviewer that heme iron can also exert oxidative stress, and this statement is revised in the manuscript.

3) In one place, ferroportin is misspelled as "feroportin".

We apologize for the mistake. In the revised manuscript, this has been corrected.

4) Means should be shown on the graph for Figure 2 A and B.

As we responded in the “Essential Revision-3”, means and SEM have been included in the revised Figure 2A and B.

Reviewer #3 (Recommendations for the authors):The manuscript by Sato et al. addresses an important question related to brain iron accumulation with aging. This is a timely question because iron accumulates in the brains of aged individuals and is believed to contribute to neurodegenerative diseases such as AD and PD.However, the data are not clear, and do not support the major conclusion that local hepcidin is responsible for iron accumulation in the brain. Some examples are given below:1. In Figure 1, the ferrozine method used to quantitate iron levels also measures iron released from ferritin. Iron sequestered in ferritin is in the ferric form, and not redox-active. Levels of ferritin in the brain increase with ageing, which is well-known, and could contribute to the observed increase in iron with aging noted by the authors.

Thank you for this comment. As the reviewer mentioned, our iron measurements may include ferritin-binding iron. However, our data showed that in addition to the cytoplasmic iron, mitochondrial iron (which is directly associated with cellular damage) is increased in the brain with aging, suggesting elevated total cellular iron in the aged brain. In addition, the activity of aconitase in both cytosol and mitochondria was decreased, suggesting the involvement of iron-mediated oxidative stress in the aged brain.

2. The mitochondrial fraction was not tested for purity. Mitochondrial markers should be evaluated and compared with the cytosolic fraction to determine the efficacy of the procedure.

As mentioned in our response to the “Essential Revision-2” and the reviewer’s comment of #2-additional limitation-3, we have confirmed the purity of mitochondrial isolation as shown in the revised Figure 1G.

3. In Figure 2 A and B, the fold change in mRNA expression is not as accurate as qPCR analysis. In the methods section the authors mention that qPCR was used. In that case, the results should be presented as ΔΔCt instead of mRNA expression. Moreover, the change in mRNA expression of HAMP1 is marginal given the standard deviation.

Since we assessed many genes associated with cellular iron regulation, we believe that showing the relative change is a better way to demonstrate the differences in their expression. We also provide the original data including CT values in the supplementary tables that readers can access. We acknowledge that the mRNA level of hepcidin in the aged brain, but not in the aged liver, showed increased SD, but the difference was statistically significant. We also confirmed that hepcidin protein expression is increased in Western blot and immunohistochemistry experiments.

4. In Figure 2 C, the hepcidin band is not clear. In the raw data for Figure 2, it seems that recombinant hepcidin peptide was fractionated in parallel in the last lane, though it is not labeled. It is unclear why the bands marked in the last lane of the 'middle exposure' are absent in the overnight longer exposure.

We appreciate the careful review of our data. The last lane was just a ladder and marked by a pen (3-, 6- and 10-kDa). The second lane form the right was lysate from the liver tissue, which is the major organ for hepcidin production.

5. The brain homogenate should be depleted of capillary endothelial cells that synthesize hepcidin, or appropriate controls need to be used to remove their contribution.

Thank you for this comment. As mentioned in our response to comments of #1-3 and #2-1, we acknowledge the limitation that we cannot reveal cell-specificity of hepcidin expression in the aged brain. This limitation has been included in the revised manuscript.

6. In the immunostained brain sample in Figure 2 E, a positive control, perhaps a section of the liver from iron depleted and iron overloaded mouse would provide confidence that the reaction is from hepcidin, though it could be hepcidin from the peripheral circulation.

We appreciate your careful review of our data. Our data showed that protein and mRNA levels of hepcidin are increased in the brain of aged mice compared to young mice, suggesting that brainderived hepcidin is actually up-regulated with aging. Therefore, the possibility that contamination of circulating hepcidin affected our analysis would be quite low. We hope the Reviewer agrees with no additional experiments, and that our current data are sufficient to show increased hepcidin expression in the aged brain.

7. In Figure 3 A, TfR1 is expected to decrease if the cells are iron loaded. The authors need to discuss this discrepancy.

Thank you for this comment. As we responded in the “Essential Revision-4”, our possible explanation for the unchanged TfR1 levels in spite of iron accumulation in the aged brain has been added to the Discussion section of the revised manuscript.

[Editors' note: further revisions were suggested prior to acceptance, as described below.]

Essential revisions:The reviewers appreciate the much improved revision of this manuscript. Several additional components need to be addressed.1) Findings of significantly decreased FPN in the western from this outbred strain with N=3 biological replicates need to be repeated and confirmed. In lieu of a repeated experiment, the authors may choose to add to the Discussion section on this point that "additional experiments are needed to confirm these findings in a larger cohort of mice of both genders."

Thank you for this great suggestion. We have added the statement of "Since the results of this study are based on a small number of immunoblot samples using single-gender of mice, additional studies are needed to confirm these findings in a larger cohort of mice of both genders" at the end of 2^nd^ paragraph of the Discussion section in the revised manuscript.

2) Given the authors' response regarding the selection of the frontal cortex as the main site of analysis, the Introduction is no longer relevant and should be edited to fall in line with the current focus, not focused on the substantia nigra and striatum.

We appreciate this valuable suggestion. According to the suggestion, we edited the introduction section. Especially, we removed the sentences mentioning Alzheimer's and Parkinson's diseases and substantia nigra and striatum and added the sentence mentioning the reason why we focused on the brain cortex in the present study. As a result, the four paragraphs in the previous introduction section have been combined into three in the revised manuscript.

Reviewer #1 (Recommendations for the authors):The authors have substantively responded to most of the reviewers' comments. Several elements remain to be resolved.1) The authors suggest in their response that they focused on the frontal cortex due to prior evidence of the importance of cortical atrophy with age. However, the introduction has a very different read and in fact states that the "substantia nigra and stratum" (misspelled striatum?), subcortical structures, are central to disease pathophysiology. Please edit the introduction more in line with the selected area of the brain studied in the current manuscript and to remove the substantive delineation of Alzheimer's and Parkinson's diseases which are NOT the focus of the current manuscript.

Thank you for this valuable suggestion. We have revised the introduction section by removing the sentences mentioning Alzheimer's and Parkinson's diseases and substantia nigra and striatum and added the sentence mentioning the reason why we focused on the brain cortex in the present study.

2) Change the title to "Aging is associated with increased brain iron through cortex-derived hepcidin expression."

We agree with the reviewer’s suggestion and the title has been changed in the revised manuscript.

3) Please spell out HPRT1 and SDCF in the methods section.

We thank for the comment, and now this is amended.

4) Discussion section regarding TFR1 regulation: please change "HIF-medicated" to "HIF-mediated".

We apologize for this typographical errors. All of "medicated" words have also been changed to “mediated” in the revised version.

5) Please add the following sentence to the Discussion section regarding TFR1, at the end: "Our data therefore cannot resolve whether TFR1 is central to age-mediated iron accumulation in the brain and if so, by what mechanism. Additional studies are necessary to clarify this important point." Alternatively, please measure other pathways that regulate TFR1.

Thank you for this excellent suggestion. The sentences suggested by the reviewer have been added to the end of 4^th^ paragraph of the Discussion section in the revised manuscript.

6) Aconitase is involved in iron responsiveness regulation in other tissues (Bullock Blood 2010). As a consequence, using aconitase as a read out of oxidative stress when iron is altered between the 2 groups is insufficient. Please remove the phrase "indicating increased oxidative stress" in the last sentence of the Results section.

According to the reviewer’s comment, we have removed the phrase “indicating increased oxidative stress” in the revised manuscript.

7) Please add "Further studies… increased TFR2 in the aged brain contributes to increased brain-derived hepcidin expression or alternatively contributes to iron accumulation with age." to the TFR2 part of the Discussion section.

Thank you for this suggestion. This is now corrected in the paper.

8) Question about muscle was intended to understand how data in muscle is relevant in a paper about the brain. This remains unclear.

As we mentioned in the introduction section, whether iron accumulation occurs in all organs or is specific to one organ was unclear despite aging has been shown to be associated with iron accumulation in various organs and tissues. Thus, we assessed aging-associated iron changes not only in the liver and the brain, but also in the skeletal muscle in the present study. We hope that the reviewer agrees that we show the iron data in the skeletal muscle in this paper.

9) Please add components of mouse strain and gender and haematologica reference for iron diet to the methods section.

We have added the information suggested by the reviewer in the method section of the revised manuscript.

Reviewer #2 (Recommendations for the authors):The authors have addressed a number of concerns, including identifying the region of brain studied and validating the enriched mitochondrial fraction. Identifying the cell types with increased Hamp1/Hepc mRNA and protein and decreased Fpn protein is important, but this has not been done. In addition, the important finding of decreased Fpn is shown in a Western with an N of only 3. This is inadequate, especially with an outbred strain and documented high variability in Hamp1/Hepc levels in Figure 2. The N for the Fpn Western should be increased, and should be correlated with Hamp1/Hepc levels.

We appreciate the reviewer’s careful concern about the limitations of our experimental results. We acknowledge that the experiments proposed by the reviewer are important. However, due to the limited amount of sample and the time it takes to age these mice, it is unfortunately difficult to perform the additional experiments or analyses that are suggested by the reviewer. Instead, as recommended by the editor, the statement of "Since the results of this study are based on a small number of immunoblot samples using single-gender of mice, additional studies are needed to confirm these findings in a larger cohort of mice of both genders" has been added to the 2^nd^ paragraph of the Discussion section in the revised manuscript as the limitation. We hope that the reviewer would agree with our response.